# Addition of a Viral Immunomodulatory Domain to Etanercept Generates a Bifunctional Chemokine and TNF Inhibitor

**DOI:** 10.3390/jcm9010025

**Published:** 2019-12-20

**Authors:** Alí Alejo, Carolina Sánchez, Sylvie Amu, Padraic G. Fallon, Antonio Alcamí

**Affiliations:** 1Centro de Investigación en Sanidad Animal, Instituto Nacional de Investigación y Tecnología Agraria y Alimentaria, Valdeolmos, 28130 Madrid, Spain; 2Centro de Biología Molecular Severo Ochoa (Consejo Superior de Investigaciones Científicas and Universidad Autónoma de Madrid), Cantoblanco, 28049 Madrid, Spain; carolina_s@cbm.csic.es; 3Trinity Biomedical Sciences Institute, Trinity College Dublin, D02 Dublin 2, Ireland; sylvie.amu@ucc.ie (S.A.); PFALLON@tcd.ie (P.G.F.)

**Keywords:** tumor necrosis factor, chemokine, receptor, inflammation, immune evasion, virus

## Abstract

The inhibition of tumor necrosis factor (TNF) through the use of either antibodies or soluble receptors is a highly effective strategy for the clinical control of chronic inflammatory conditions such as rheumatoid arthritis. Different viruses have similarly exploited this concept by expressing a set of specifically tailored secreted TNF decoy receptors to block host inflammatory responses. Poxviruses have been shown to encode at least two distinct molecules, termed Cytokine response modifier D (CrmD) and CrmB, in which a TNF inhibitor is combined with a chemokine inhibitor on the same molecule. The ectromelia virus CrmD protein was found to be a critical determinant of virulence in vivo, being able to control local inflammation to allow further viral spread and the establishment of a lethal infection. Strikingly, both the TNF and the chemokine inhibitory domains are required for the full activity of CrmD, suggesting a model in which inhibition of TNF is supported by the concomitant blockade of a reduced set of chemokines. Inspired by this model, we reasoned that a similar strategy could be applied to modify the clinically used human TNF receptor (etanercept), producing a generation of novel, more effective therapeutic agents. Here we show the analysis of a set of fusion proteins derived from etanercept by addition of a viral chemokine-binding protein. A bifunctional inhibitor capable of binding to and blocking the activity of TNF as well as a set of chemokines is generated that is active in the prevention of arthritis in a murine disease model.

## 1. Introduction

In an ongoing evolutionary arms race, viruses have contributed to shape the innate and adaptive immune system of their vertebrate hosts through the development of effective countermeasures directed against it. Thus, different viruses deploy sets of proteins targeted at distinct host response pathways depending on their target tissue as well as their replication strategies [1,2,3]. Common elements frequently targeted by various viruses and employing different strategies include the antiviral interferon IFN response, the inflammatory response or the antigen presentation pathways [2]. A relevant strategy found mostly in different families of large dsDNA viruses such as herpesviruses and poxviruses is the use of viral secreted proteins targeting cytokines, the soluble mediators of the immune response [4]. These viral proteins in general act by binding to their ligands, thus blocking the recognition and/or activation of their cognate receptors. 

Indeed, this strategy has been successfully adapted for its use in the clinic through the development of blocking monoclonal antibodies, modified versions of cellular receptors inhibiting various cytokines or antagonist cytokine ligands that compete for binding of the agonist ligands to their receptor. This is best exemplified for the case of clinical inhibition of interleukin-1(IL-1), for which three different approved biologics exist, namely the IL-1 receptor antagonist anakinra, the soluble decoy receptor rilonacept, and the neutralizing monoclonal antibody canakinumab [5]. These are successfully used for the treatment of several inflammatory conditions including rheumatoid arthritis (RA), ankylosing spondylitis, Cryopyrin-associated periodic syndromes, or Familial Mediterranenan fever, among many others [6]. Other cytokines targeted include IL5, IL6, or receptor activator of nuclear factor κ B ligand (RANKL), while new ones are under study and development [7,8,9,10,11]. Among the most successful in this pharmacologic group of biological treatments are those directed against tumor necrosis factor (TNF)-induced signaling, which are currently in use for the treatment of different chronic inflammatory conditions, with RA being one of their major indications [12,13]. Currently, five different inhibitors of TNF are in use. Infliximab, adalimumab, and golimumab are monoclonal antibodies against TNF, and certolizumab is a polyetyhylene glycol-linked anti-TNF antigen-binding fragment (Fab’), while etanercept is a recombinant fusion protein including the extracellular domain of human TNF receptor 2 (TNFR2) fused to the human immunoglobulin G1 fragment crystallizable (IgG1Fc) domain. 

Interestingly, poxviruses have developed a set of secreted TNF inhibitors that can be divided into two groups. On one hand, the high-affinity inhibitor of human TNF encoded by Tanapox virus 2L protein shows no homology to cellular TNF receptors. It has been suggested that this protein may have been adapted to this function through extensive sequence modification over evolutionary times [14]. An orthologue of this protein is found exclusively in other members of the genus yatapoxvirus and in swinepox and deerpox virus. On the other hand is the more generally distributed family of secreted proteins with homology to the cellular TNF receptor superfamily 2 (TNFRSF2) termed viral TNF receptors (vTNFRs) that encompasses four distinct proteins termed Cytokine response modifier B (CrmB), CrmC, CrmD, and CrmE. Collectively, vTNFRs can bind to and inhibit the activities of TNF alpha (TNF) in either its secreted or transmembrane forms, lymphotoxin alpha (LTα) and lymphotoxin beta (LTβ), and their specific binding affinities and activities have been well characterized [15,16]. Both CrmB and CrmD proteins include a C-terminal extension with no sequence similarity to the TNFRs, which was found to correspond to a chemokine-binding domain and was termed the smallpox virus-encoded chemokine receptor (SECRET) domain [17]. This domain can bind to a limited set of human chemokines (CK) of all four different families (CCL20, CCL25, CCL28, CXCL12b, CXCL13, CXCL14, XCL1, and CX3CL1) and it has been shown to adopt a compacted beta sandwich fold reminiscent of other viral chemokine-binding proteins that accounts for its specific binding characteristics [18]. Moreover, the SECRET domain has been shown to be able to inhibit CK-induced cell migration by blocking both receptor and glycosaminoglycan GAG binding sites [18]. While the SECRET domain can be found independently in three different viral proteins, it was shown to act in concert with the TNF-binding moieties in CrmD and CrmB, which are thus effectively bispecific cytokine inhibitors. Importantly, in vivo experiments using the mousepox model showed that both TNF- and CK-blocking activities are important determinants of virulence and have complementary roles in the control of local inflammation during infection [19].

In inflammation-driven disease, chemokines play a central role by mediating cell migration and have, therefore, long been considered a critical target for the development of new therapies [20]. Guided by our previously described observations of the poxviral model, we hypothesized that addition of a viral chemokine inhibitor such as the SECRET domain to currently in use inhibitors of TNF could generate more effective biologicals for the treatment of inflammatory conditions. In initial experiments used as proof of concept, we found that the CK-binding activity could be transferred to other viral TNFRs by fusing the CrmB-derived SECRET domain to them, although these constructs showed a reduced binding affinity for their CK ligands. Here we describe the construction of chimeric constructs containing the hTNFR2 currently used in the clinic fused to SECRET domains of different origin and nature. A bifunctional molecule capable of effectively blocking both TNF and CK ligands in vivo is described.

## 2. Materials and Methods

### 2.1. Cells and Reagents

The L929 murine fibroblast cell line was grown in Dulbecco´s modified Eagle´s medium supplemented with 10% fetal calf serum (FCS). The human T lymphoblast cell line MOLT-4 was grown in RPMI 1640 medium supplemented with 10% FCS. The *Trichoplusia ni*-derived insect cell line Hi5 (Thermofisher Scientific, Waltham, Mass, USA) was grown in TC-100 medium supplemented with 10% FCS for adherent cell culture and in ExpressFive serum-free medium for suspension culture and protein expression experiments. Cytokines were purchased from R&D Systems (Minneapolis, MN, USA).

### 2.2. Cloning of hTNFR2 Fusion Constructs and Generation of Recombinant Baculoviruses

Construction of plasmids pMS40 (pBac1-CrmD-Fc) and pRM6 (pFastBacMEL-hsTNFR2-Fc) and the generation of the corresponding baculoviruses for the expression of ectromelia virus (ECTV) CrmD-Fc and hTNFR2–Fc proteins have been described before [15,21]. To obtain all additional plasmids, the sequences corresponding to the indicated domains were PCR-amplified using appropriate oligonucleotides and templates as required. The purified fragments were subcloned using the same strategy into an EcoRI/NotI digested intermediate cloning plasmid to place the different SECRET domain coding sequences immediately 3’ the hTNFR2 and 5’ to the human IgG1 coding sequences. The complete coding sequence of all plasmids was verified for absence of mutation by sequencing, and full information on the cloning strategy and final sequence of the plasmids and encoded fusion proteins is available upon request from the authors. Baculoviruses were generated by transfection of a recombined bacmid into Hi5 cells using the Bac-to-Bac Baculovirus expression system (Invitrogen) according to the manufacturer’s instructions. High-titer baculovirus stocks were stored at 4 °C and used for expression assays. 

### 2.3. Expression and Purification of Recombinant Protein Constructs

Hi5 cells grown in suspension were infected at high multiplicity (>5 pfu/cell) with the corresponding recombinant baculovirus. The supernatants were harvested at 48–72 hours post-infection depending on the construct by centrifugation and concentrated using a Minimate Tangential Flow Filtration System (PALL). The concentrates were then diafiltered into 20 mM phosphate buffer pH 7.0 before immunoaffinity purification. This was performed using prepacked Protein A-sepharose columns (HiTRap, GE Heathcare, Chicago, IL, USA) following the manufacturer’s instructions. The purified proteins were dialysed into 0.2 M Hepes, 1.5 M NaCl, pH 7.4 buffer containing 0.01% sodium azide and quantified using a BCA assay and/or densitometry of Coomassie blue-stained SDS-PAGE, on which different amounts of each protein were loaded along purified bovine serum albumin (BSA) standards (Sigma-Aldrich, Saint Louis, MO, USA). For in vivo experiments, the recombinant proteins were further purified by size exclusion chromatography on a Superdex 200 column (GE Healthcare, Chicago, IL, USA) using an AKTA Prime Plus instrument, and the fractions tested for anti TNF activity before being pooled. Purified protein stocks were kept at −70 °C. All protein samples used in vivo were tested for absence of endotoxin using the ToxinSensor™ Chromogenic LAL Endotoxin Assay Kit (GenScript, Piscataway, NJ, USA).

### 2.4. TNF-Induced Cytotoxicity Assay

Cytotoxicity assays were performed as described before [17]. Briefly, L929 cells grown on 96-well plates were incubated with 20 ng/mL (1.2 nM) of hTNF or 10–20 ng/mL (1.2 nM or 2.4 nM) of mTNF for 16 h in the presence of 4 µg/mL ActinomycinD (Sigma, Saint Louis, MO, USA) to induce cell death. Relevant recombinant proteins were preincubated at the indicated doses with TNF for 2 h at 37 °C before addition to the cell monolayers. Experiments were carried out in triplicate wells. After the incubation period, cell death was assessed using the Cell Titre Aqueous One Solution viability assay (Promega, Madison, WI, USA) according to the manufacturer´s instructions. The data were analyzed for statistical significance by performing multiple *t*-test comparisons among selected groups using Prism 6.0 software (GraphPad, San Diego, CA, USA).

### 2.5. Cell Migration Assay

Migration of MOLT-4 cells towards chemokine was assayed using 96-well ChemoTx plates with 5 µm pores (Neuro Probe, Gaithersburg, MD, USA). Chemokine (mCCL25) at 100 nM was incubated in the presence of increasing amounts of recombinant proteins, as indicated in the bottom wells, for at least 15 min at 37 °C. Cells (2.5 × 10^5^ per well) previously washed and resuspended in 0.1% FCS in RPMI1640 medium were carefully laid on the top filter and allowed to migrate for 2–6 h at 37 °C. After this period, nonmigrated cells from the top filter were rinsed away with phosphate buffered saline (PBS), and the number of migrated cells in each well determined using the Cell Titre Aqueous One Solution viability assay (Promega, Madison, WI, USA). The number of cells was extrapolated from a standard curve with known cell numbers prepared in triplicate for each assay, as suggested by the manufacturers. 

### 2.6. Surface Plasmon Resonance

The kinetic affinity constants of the hTNFR2–SCP3 fusion protein were determined using a Biacore X-100 biosensor (GE Healthcare, Chicago, IL, USA). Recombinant protein was amine coupled onto CM4 chips at a low density (about 500 response units) and increasing concentrations of analytes in the range from 2 nM to 250 nM injected in HEPES-EP buffer (GE Healthcare, Chicago, IL, USA) at 30 µL/min for 60 s. Dissociation periods of 300 s were recorded during single-cycle experiments. Final dissociation of analytes was achieved using 10 mM glycine, pH 2.0 injections for 30 s. The data were automatically processed and fitted to a 1:1 Langmuir model using BIAevaluation 2.0.1 software (GE Healthcare, Chicago, IL, USA). Bulk refractive index changes were removed by subtracting the responses recorded in the reference cell, and systematic errors were corrected by subtracting the response of a buffer injection from all sensorgrams.

### 2.7. Collagen-Induced Arthritis (CIA) Murine Model

Groups of 3–6 female 6–8 week old DBA/1OlaHsd mice (Harlan, Bicester, UK) were housed under noise-free conditions and acclimated to a once daily visit and handling regime for a two week period before the start of the experiment. On day 0 of the experiment, mice were injected intradermally at the base of the tail with 100 µl of complete Freund´s adjuvant preemulsified bovine collagen type II (Hooke Laboratories). On day 21, a second injection of bovine collagen type II, preemulsified in incomplete Freund´s adjuvant was performed. At the indicated time points, intraperitoneal injections of the recombinant proteins and doses indicated in 100 µL volume were given. Animals were examined daily for clinical signs of arthritis and scored according to a preestablished scoring table (a score from 0 to 4 was awarded for each paw, with 0 for no inflammation, 1 for swelling of one digit, 2 for swelling of all digits and paw, 3 for severe inflammation of the complete paw and digits or ankyloses, and 4 for necrosis, giving a total maximum possible score of 16 per animal). To ensure that only animals developing arthritis-like signs were considered, after cessation of treatment, animals were evaluated for an additional period of up to 10 d, and those not showing clinical scores before and after end of treatment were withdrawn from the assay. All animal experiments were performed with special efforts to minimize animal suffering, in compliance with national and international regulations, and were approved by the Ethical Review Board of Consejo Superior de Investigaciones Científicas. The data were analyzed for statistical significance by performing multiple *t*-test comparisons using Prism 6.0 software (GraphPad, San Diego, CA, USA).

## 3. Results

### 3.1. Generation of a Collection of hTNFR2-SECRET Fusion Proteins

The poxviral SECRET chemokine-binding and inhibitory domain has been found both independently in the SECRET-containing-protein 1 (SCP1), SCP2, and SCP3 as well as fused to the TNF inhibitory domains of the vTNFRs CrmD and CrmB. While all of them have been found to bind to the same set of murine and human chemokines, their potential differences in terms of affinity or inhibitory profiles have not been analyzed in detail [17]. In variola virus (VARV) CrmB, the SECRET domain including residues T194 to L348 was found to bind to the same set of chemokines as the complete VARV CrmB protein and could be fused to another two vTNFRs (CrmC and CrmE), conferring chemokine-binding ability upon them while preserving their TNF-binding capability [17]. More detailed structure–function analyses performed with ECTV CrmD have shown that an equivalent domain (N181-D320) in this protein retains the same binding properties, but requires an N-terminal extension up to the highly conserved residue F163 to maintain full binding affinities for all of its known ligands when expressed by themselves [22].

Therefore, with the aim of conferring chemokine inhibitory activity to the soluble hTNFR2–Fc receptor, we first generated a collection of hTNFR2–Fc receptors in which we inserted SECRET domains of different origin and variable length between the hTNFR2 and Fc domains, as described in Table 1. Recombinant baculoviruses to express all of these nine constructs were generated, and those six that consistently showed higher protein expression and secretion levels were selected for further characterization. The recombinant proteins were purified by immunoaffinity chromatography alongside the previously described hTNFR2–Fc construct as well as the CrmD-Fc fusion protein (Figure 1). All of the purified proteins showed single bands with the expected mobility on SDS-PAGE analysis, except for the hTNFR2–CrmB SECRET F176 construct, in which contaminating bands were detected that probably correspond to unwanted proteolytic processing products.

### 3.2. Analysis of the TNF Inhibitory Activity of the hTNFR2–SECRET–Fc Constructs

We next determined whether the purified recombinant proteins preserved the TNF inhibitory activity upon fusion of the different SECRET domains. To do so, we employed a hTNF-mediated cytotoxicity assay previously established in our laboratory in which cell death can be efficiently blocked by hTNFR2 [15]. At the doses tested, only three of the assayed constructs were able to protect to some degree against hTNF induced cytotoxicity (Figure 2). Interestingly, these were the fusions in which the SECRET domain was that corresponding to the predicted full-length secreted SCPs 1–3 as opposed to the ones derived from fragments of different lengths from the CPXV CrmB protein, suggesting that in the latter, the folding or binding ability of the hTNFR2 is compromised by the added residues.

We therefore focused on the three active constructs to perform a side-by-side comparison of their TNF inhibitory properties as compared with those of the parental hTNFR2. To do this, both murine and human TNF were tested due to their relevance for murine experimental models as well as in the clinic.

We first tested the activity of the hTNFR2-SCP2 fusion protein, which had shown the lowest hTNF inhibitory activity in the previous assay. As displayed in Figure 3A, this protein showed diminished hTNF-blocking activity and undetectable inhibitory activity against mTNF as compared with hTNFR2. Thus, while 100 ng of hTNFR2 (corresponding to a molar excess of approximately 15 fold) was sufficient to provide full protection against hTNF-induced cytotoxicity, up to 5 µg of the hTNFR2–SCP2 was needed to obtain the same effect. In the case of mTNF, which is less efficiently blocked by hTNFR2, no inhibitory activity of hTNFR2–SCP2 could be detected at any of the doses tested.

On the contrary, the fusion proteins hTNFR2–SCP1 and hTNFR2–SCP3 showed inhibitory activities against both hTNF and mTNF that were comparable to those afforded by hTNFR2 (Figure 3B). The apparent differences at the lower doses of recombinant protein tested probably stem from the difference in the molar excess of inhibitor vs. TNF, which is about 1.4 times lower in the case of the fusion proteins. Importantly, both fusion proteins achieved full protection against both mTNF and hTNF at the higher doses tested, indicating that addition of either SCP1 or SCP2 in the latter constructs did not affect the TNF inhibitory activity of hTNFR2.

### 3.3. Chemokine Inhibitory Properties of Fusion Proteins hTNFR2-SCP1 and hTNFR2-SCP3

Next, we wished to determine whether the fusion proteins had acquired the capacity to block chemokine-induced migration. To this end, we performed CCL25-induced chemotaxis assays with MOLT4 cells in the absence or the presence of the recombinant proteins. Incubation of the chemokine with increasing amounts of the full-length CrmD protein completely blocked CCL25-induced cell migration (Figure 4), as had been shown before [17]. The hTNFR2, which does not bind chemokines, did not impair cell movement in this assay. Enhanced CCL25-induced cell migration was observed in the presence of low doses of all recombinant proteins, independent of their ability to inhibit cell migration at higher doses. Relevantly, both hTNFR2-SCP1 and hTNFR2-SCP3 were able to prevent chemokine-induced migration in a dose-dependent manner, achieving complete blockade at 10–20 fold molar excess over the chemokine. This shows that fusion of either SCP confers chemokine inhibitory activity to hTNFR2, generating effectively bifunctional TNF and chemokine inhibitory molecules. Because full blockade of cell migration was obtained in the presence of a lower molar excess of hTNFR2-SCP3 as compared with that of hTNFR2-SCP1, the former recombinant protein was selected for further characterization. 

### 3.4. The Fusion Protein hTNFR2-SCP3 Binds TNF and Chemokines with High Affinity

We then determined the kinetic affinity parameters of the interaction of hTNFR2-SCP3 with different ligands using surface plasmon resonance. As shown on Table 2, hTNFR2-SCP3 interacted with high affinity with hTNF, showing both association and dissociation parameters similar to those calculated before for hTNFR2. Additionally, high-affinity binding in the nanomolar range was calculated for three different chemokine ligands of the SECRET domain. While no affinity data for SCP-3 have been published, the calculated affinities were comparable to those calculated for the SECRET domain of CrmD either on its own or in the context of the full-length protein [22]. Overall, the data suggest that the hTNFR2-SCP3 fusion protein may act as an efficient TNF and CK inhibitor in vivo.

### 3.5. The hTNFR2-SCP3 Fusion Protein Can Delay the Development Clinical Signs Associated with Arthritis in a Murine Model

RA is one of the major indications for etanercept in the clinic. We therefore addressed the capacity of the newly developed version targeting chemokines as well as TNF to modify the course of the disease as compared with the conventional biological. To do so, we employed the well-established CIA model [23]. After the second immunization with collagen, DBA/J mice treated with vehicle only developed typical signs of arthritis, including apparent joint swelling that increased with time (Figure 5). When treated with intraperitoneal injections of 50 µg or 100 µg of ENBREL (the commercial name of etanercept) every three days, development of clinical signs was reduced over the time course of treatment in a dose-dependent manner, as expected, with the dose of 100 µg showing statistically significant (*p* < 0.01) differences as compared with the untreated control group from day 28 to the end of the experiment. The 50 µg dose of hTNFR2-SCP3 injected in the same way produced an effect similar to that of 100 µg of ENBREL, with statistically significant differences to the control group, as before. Moreover, no significant differences were detected when the ENBREL 100 µg and hTNFR2-SCP3 50 µg treatment groups were compared, while the latter was significantly better at reducing clinical signs than the ENBREL 50 µg from day 28 of the experiment on (Figure 5A). This shows that the modified recombinant protein can block development of signs of illness in the golden model for this pathology and it suggests that it may be more efficient than the commercial biological, at least in terms of the dose required to achieve a significant reduction of clinical scores. In parallel, treatment with purified CrmD protein, the viral inhibitor of both TNF and chemokines, was also capable of attenuating the course of disease (Figure 5A). In a repeat experiment, a 50 µg dosing of hTNFR2-SCP3 showed a similar capacity to impair clinical signs as 100 µg ENBREL or 50 µg of in-house-produced hTNFR2. Again, both ENBREL and hTNFR2-SCP3 significantly reduced the development of clinical signs for longer periods (Figure 5B), while no significant differences were detected between both regimes at any time point of the experiment. Altogether, these experiments showed that the newly developed hTNFR2-SCP3 fusion protein has a similar or increased capacity to block disease development in the CIA mouse model as hTNFR2, indicating that addition of a functional chemokine inhibitory domain may augment the efficacy in terms of the parameters determined here of the widely used etanercept molecule.

## 4. Discussion

In this report, we have shown that the TNFR etanercept can be modified to include a chemokine inhibitory function without compromising its primary clinical efficacy in a murine model of RA. This was done by adding a previously identified viral chemokine-binding domain to the recombinant protein, generating a novel, bifunctional fusion construct. In a related approach, a dual inhibitor targeting simultaneously TNF and IL-17 has been tested and shown to be effective in reducing pathology in RA and psoriasis murine models [24,25]. However, in this case an IL-17 single-chain, variable fragment was fused to a soluble TNFR1 receptor, as opposed to the TNFR2 receptor of etanercept, confounding possible comparisons due to differences in binding specificity for TNF ligands of the human receptors [26].

Although the generic structure of the SECRET chemokine-binding domain employed here is known [18], there is currently no structural information as to how this domain folds in the context of the viral dual TNF and chemokine inhibitor CrmB, where it was first identified, nor of the possible variations of the SECRET domain found in the different SCPs. Therefore, we employed an unbiased approach by fusing different SECRET domain-containing constructs to the hTNFR2 in etanercept. This was done maintaining the same domain architecture as present on the viral bifunctional receptors CrmD and CrmB, that is, an N-terminal TNF-binding domain, followed by a CK-binding domain. The IgG1 Fc portion was located at the C-terminal end of the fusion molecules, as in etanercept. The fusion of the hTNFR2 to SECRET domains derived from CrmD did not result in the production of readily purifiable constructs, which may be related to a reduced expression or stability of these fusion proteins. Fusion of the hTNFR2 to SECRET domains from CrmB was possible, although the purified recombinant proteins showed a compromised TNF inhibitory activity. The sequence similarity of the poxvirus SECRET domains is relatively low (around 30%), and it is somehow surprising that fusion of the hTNFR2 to the SECRET domains from CrmB and CrmD, which are naturally fused to the viral TNF-binding domain, affect the stability and/or TNF-binding capacity of the human receptor. This may be due to a compromised folding of the TNF-binding domain or a steric hindrance of this function as a result of inclusion of the novel domain. Fusion of the hTNFR2 to SCPs, the SECRET domains expressed as independent proteins, gave better results, with two constructs retaining TNF-binding activity. Moreover, at least one of them showed high affinities for its chemokine ligands, which were equivalent to those described for these viral chemokine-binding proteins (vCKBPs) [22], suggesting that both structural domains are correctly folded in this case. While the SECRET domain is thought to function as a monomer [18], Fc fusion proteins are known to tend to form dimers, raising the question as to how this can possibly affect interaction with CKs. To understand how both binding and inhibitory activities are integrated in a single molecule, further structural characterization of this construct, as well as the dual viral receptors CrmB and CrmD will be of interest. The oligomerization status of these molecules as well as the binding stoichiometries to their different ligands remain to be addressed. This might, in turn, facilitate the rational design, in the future, of more effective dual inhibitors. Interestingly, we found that fusion of a distinct vCKBP (protein 35K, a.k.a. vCCI), which has a different chemokine-binding profile, was also possible, suggesting that a specific tailoring of these novel dual receptors could provide a range of chemokine inhibitory profiles for use in different settings. Thus, the known set of vCKBPs or the tick evasin family [27] could be used for this purpose.

Chemokines modulate the movement of different cell types and populations into and out of the sites of affection in all chronic inflammatory conditions. Because these cells are in large part responsible for the pathological manifestations of these diseases, the clinical regulation of their movement is a long sought-after but as yet unrealized therapeutic target for their treatment [20]. Chemokines are a large group of related cytokines mediating cell movement that act by interacting with their cognate chemokine receptors [28]. The pleiotropy of this signaling system as well as the failure to design highly efficient antagonists to regulate it have hindered the development of effective drugs to modulate it [29]. By contrast, vCKBPs are efficient immunomodulators that have been shown to be able to control cell migration and influence pathology in different contexts [30]. For example, the secreted herpesviral M3 protein, which can bind to chemokines from all four different chemokine families [31,32] is involved in the development of infectious mononucleosis during MHV-68 infection of mice and is thought to block CD8 T cell recruitment into lymphoid tissue [33]. Relevantly, vCKBPs have been assessed as potential therapeutic inhibitors of chemokine activity in different models of disease. Thus, expression of the poxviral CC-chemokine-binding protein 35K from a recombinant adenovirus impaired macrophage migration in a murine atherosclerosis model, significantly reducing the observed lesion area [34]. Using a similar approach, inflammatory-induced but not physiological angiogenesis was suppressed in vivo by conditionally modulating the expression of angiogenetic factors VEGF and HIF1-α [35]. Administration of a recombinant vCCI-Fc protein in a CIA model resulted in reduced infiltration of inflammatory cells into knee joints as well as limited cartilage degradation, possibly as a result of the observed blockade of the egress of IFN-γ-secreting and activated T-cells from the spleen towards the inflamed tissues [36]. Serin protease inhibitors (Serpins) encoded by poxviruses have shown potent anti-inflammatory activity in a number of disease models [37]. The M-T7 vCKBP encoded by myxoma virus, in combination with viral Serpin-1, reduces the early inflammatory response after spinal cord injury [38].

Blockade of TNF by soluble TNF inhibitors including receptors and different monoclonal antibodies is an effective therapeutic strategy widely used for the treatment of RA and other inflammatory conditions [39]. During viral infection, concomitant blockade of both TNF and a reduced set of chemokines by a single secreted protein was found to completely abrogate local inflammation in the mousepox intradermal infection model, effectively ablating the host response to produce a lethal infection [19]. Strikingly, both TNF and chemokine inhibition were found to be required for this effect, which may reflect the local crosstalk between both signaling pathways as well as the cell types that produce and respond to them. The role of specific chemokines in this context has not yet been elucidated, although about a third of infiltrating lymphocytes detected in the absence of CrmD expressed CCR10, the receptor for CCL27 and CCL28, both of which are bound by this dual TNF- and CK-binding protein. In a transgenic mouse model of Crohn´s-like inflammatory bowel disease, expression of CrmD in intestinal epithelial cells prevented disease development. Inhibition of TNF-induced chemokine expression, as well as direct, local inhibition of CXCL13 induced B-cell migration both contributed to the reduced leukocytic infiltrates observed in the ileum of affected animals [40]. We expect that the recombinant hTNFR2-SCP3 can act on chemokines both indirectly, through its effect on TNF, but also directly, by blocking chemokine activity in situ. In this sense, it is worth noting that beyond their direct effects on cell migration, chemokines also mediate other processes including neovascularization or leukocyte activation, among others [28,41]. In RA, for example, CXCL12, which is thought to be important for the recruitment of CD4^+^ memory T cells into the synovium [42], can also activate fibroblast-like synoviocytes, upregulating CXCL8 expression [43] and accounts for some of the characteristic angiogenetic activity [44]. Other chemokines directly targeted by the SCP3 (CCL20, CCL25, CCL28, CXCL12b, CXCL13, CXCL14, XCL1 and CX3CL1) have been shown to contribute to the pathogenesis of RA, including CXCL13, which may recruit B-cells into inflamed joints, or CCL28 that participates in angiogenesis [45]. Moreover, inhibition of CXCL13 with a neutralizing antibody ameliorated signs of RA in the murine CIA model [46], as did blockade of CXCL12 [47]. CX3CL1 is involved in the migration of synovial fibroblasts and in angiogenesis, and its blockade using specific antibodies was found to reduce inflammatory cell infiltration and bone erosion in a murine RA model [48]. CCL20 may have an important role in RA development [49], and its blockade has been shown to hinder migration of Th17 cells to the inflamed joints in a murine model [50]. Likewise, CCL25-mediated cell migration and production of inflammatory mediators may contribute to RA [51]. Thus, hTNFR2-SCP3 can block several distinct chemokines relevant for RA development simultaneously, which may represent an advantage over other, more specific inhibitors. A more profound characterization of the hTNFR2-SCP3 treated CIA model at the histopathological and cellular level will contribute to a better understanding of the possible mechanisms of action of this novel recombinant protein. Etanercept-based dual TNF and chemokine inhibitors might be a relevant treatment options for many of the TNF-driven inflammatory diseases.

## Figures and Tables

**Figure 1 jcm-09-00025-f001:**
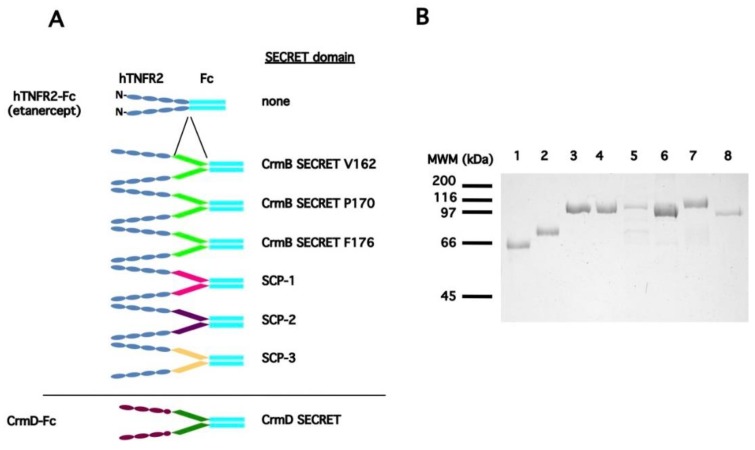
Purification of hTNFR2–SECRET–Fc fusion protein constructs. (**A**) Shows a schematic depiction of the fusion protein constructs obtained as well as the nature of their SECRET domains. As a reference, the schematized domain architecture of the recombinant CrmD-Fc fusion protein is shown below. The ovals represent TNFR characteristic cysteine-rich domains, while the SECRET domains are presented as parallelograms. Colors are used to indicate the different origin of the domains. The IgG1 Fc domains are shown as cyan-colored straight bars in all cases. (**B**) Coomassie blue-stained SDS-PAGE analysis of the IgG immunoaffinity purified constructs is as follows: lane 1: hTNFR2–Fc; lane 2: CrmD–Fc; lane 3: hTNFR2–CrmB SECRET V162–Fc; lane 4: hTNFR2–CrmB SECRET P170-Fc; lane 5: hTNFR2–CrmB SECRET F176-Fc; lane 6: hTNFR2–SCP1–Fc; lane 7: hTNFR2–SCP2–Fc; lane 8: hTNFR2-SCP3-Fc. MWM positions are indicated on the left. Abbreviations: kDa, kilodalton.

**Figure 2 jcm-09-00025-f002:**
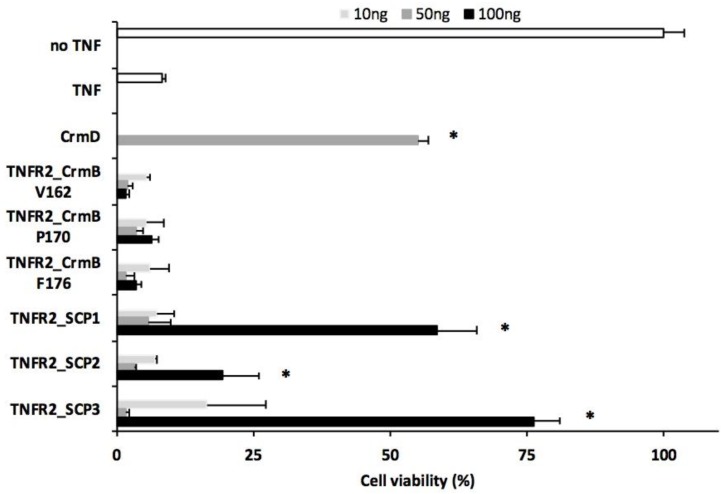
Inhibition of hTNF-induced citotoxicity by hTNFR2-SECRET fusion proteins. L929 cells were incubated with hTNF for 16 h to induce cell death. The hTNF was preincubated with increasing doses of each recombinant protein as indicated before adding it to the monolayer. As controls, cells without TNF (no TNF), without recombinant protein (TNF), or treated with TNF that had been preincubated with recombinant CrmD protein are shown. Cell viability was determined after the incubation period in triplicate wells for each condition, and mean data + SD referring to 100% viability in the no TNF sample is shown. Asterisks indicate statistically significant (*p* < 0.01) differences detected between hTNFR2_SCP fusion proteins and the control hTNF only group. Abbreviations: SD, standard deviation.

**Figure 3 jcm-09-00025-f003:**
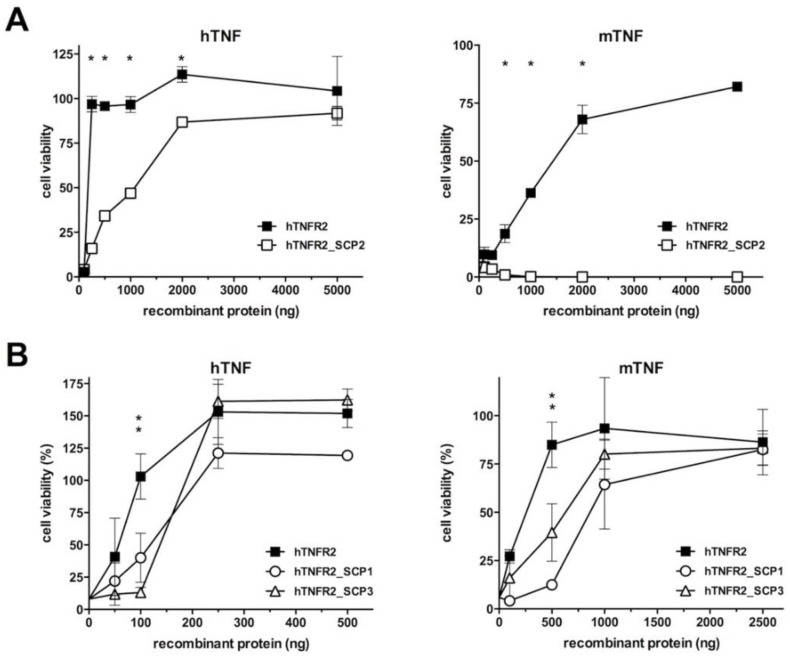
Inhibitory activity of hTNFR2-SCP1, hTNFR2-SCP2, and hTNFR2-SCP3 proteins in comparison with hTNFR2. Cytotoxicity assays using hTNF and mTNF as indicated on each panel were carried out on L929 cells, which were incubated in the absence or the presence of increasing amounts of recombinant proteins for 16 h. After that period, cell viability was determined in triplicate wells for each condition, and mean data ± SD referring to 100% viability in the no TNF sample is shown in all panels. Comparisons of hTNFR2 to hTNFR2_SCP2 (**A**) or hTNFR2_SCP1 and hTNFR2_SCP3 (**B**) are shown. Asterisks indicate statistically significant (*p* < 0.01) differences detected between hTNFR2_SCP2 fusion protein and the control hTNFR2 protein (A, *) or between hTNFR2_SCP1 and hTNFR2_SCP3 fusion proteins and the control hTNFR2 protein (B, **).

**Figure 4 jcm-09-00025-f004:**
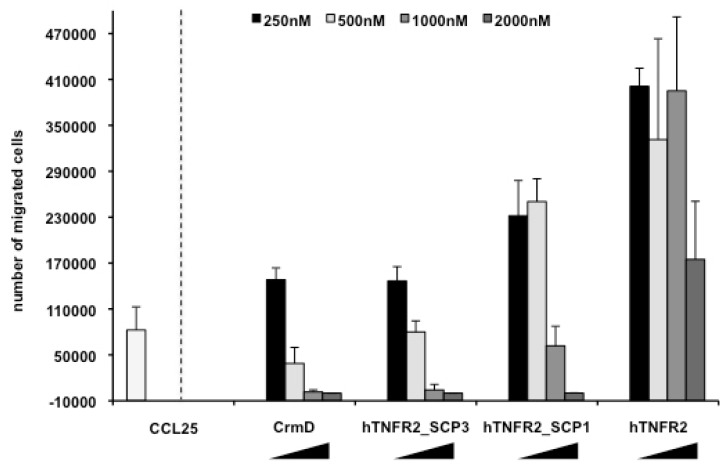
Chemokine activity inhibition by the recombinant hTNFR2-SCP1 and hTNFR2-SCP3 proteins. MOLT-4 cells were subjected to mCCL25-induced migration on Transwell filter plates in the absence or presence of increasing doses of hTNFR2-SCP1 and hTNFR2-SCP3 as compared with hTNFR2 as indicated. The known chemokine inhibitory protein CrmD was included in the assay as a positive control. The number of migrated cells was determined in triplicate wells, and mean + SD for each condition is shown.

**Figure 5 jcm-09-00025-f005:**
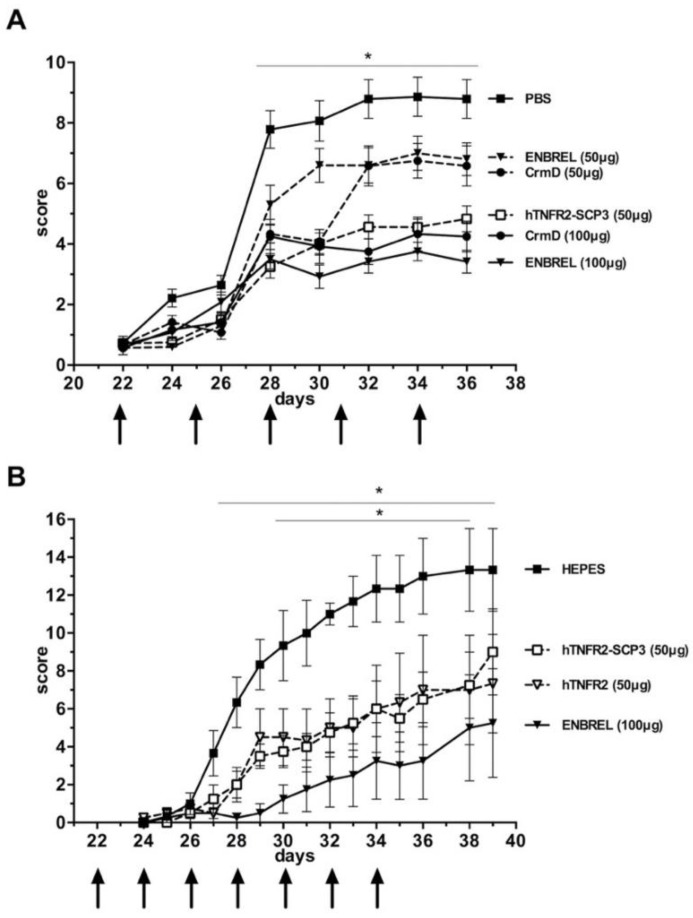
The hTNFR2-SCP3 protein prevents the development of clinical signs in a murine arthritis model. Development of arthritis was induced in groups of DBA/J mice by two injections of Freund´s adjuvant emulsified collagen separated by 21 days. On the day after the second injection (day 22 of the experiment), treatment by intraperitoneal injection (arrows) with vehicle (PBS/HEPES) or the indicated recombinant proteins was initiated. ENBREL refers to the commercial preparation of etanercept, while hTNFR2 is an in-house purified version of the same protein, as stated in the Materials and Methods section. In panels (**A**) and (**B**), two independent experiments are shown using slightly different treatment groups (*n* = 6 in (**A**) and *n* = 4 in (**B**), except *n* = 3 for the HEPES control) and dosage regimes, as indicated. Clinical scores were determined according to a standard scoring table and mean +/− SD is represented for each time point analyzed. The asterisk and bar on panel (**A**) indicate time points at which statistically significant (*p* < 0.01) differences occurred between the ENBREL 50 µg and hTNFR_SCP3 50 µg treatment groups. In panel (**B**), asterisks and bars indicate significant differences detected between the ENBREL 100 µg (upper bar) or the hTNFR_SCP3 50 µg (lower bar) as compared with the untreated control group.

**Table 1 jcm-09-00025-t001:** Recombinant Fc fusion proteins used in this report.

^a^ Protein Expressed	^b^ SECRET Domain	Predicted pI/MWof the Secreted Protein
CrmD	na	5.73/60053.42
hTNFR2	none	6.89/52797.53
hTNFR2-CrmD SECRET N181 *	ECTV_CrmD N181-D320	5.74/68594.08
hTNFR2-CrmD SECRET P153 *	ECTV_CrmD P153-D320	5.62/71693.57
hTNFR2-CrmD SECRET F163 *	ECTV_CrmD F163-D320	5.68/70650.41
hTNFR2-CrmB SECRET V162	CPXV_CrmB V162-L355	5.80/74763.68
hTNFR2-CrmB SECRET P170	CPXV_CrmB P170-L355	5.86/73915.75
hTNFR2-CrmB SECRET F176	CPXV_CrmB F176-L355	5.86/73320.09
hTNFR2-SCP1	CPXV_V218 S19-G193	6.16/73188.66
hTNFR-SCP2	ECTV_E12 N22-N202	5.54/73124.16
hTNFR-SCP3	ECTV_E184 Y18-F181	6.22/72356.33

^a^ Names of the recombinant Fc-fusion proteins expressed. ^b^ Origin of the fused chemokine inhibitory SECRET domains and the respective amino acid positions. * Indicates poorly expressed constructs that were not further characterized. Abbreviations: CPXV, cowpox virus; Crm, cytokine response modifier; ECTV, ectromelia virus; Fc, fragment crystallizable; hTNFR2, human tumor necrosis factor receptor 2; MW, molecular weight; pI, isoelectric point; SCP, SECRET-containing protein; SECRET: smallpox virus-encoded chemokine receptor.

**Table 2 jcm-09-00025-t002:** Calculated kinetic affinity parameters for the interaction of hTNFR2–SCP3 with selected ligands.

	hTNFR2_SCP3			^a^ CrmD	^a^ hTNFR2
Ligand	k_a_ ^b^ ± SE × 10^5^ (1/Ms)	k_d_ ^c^ ± SE × 10^−3^ (1/s)	K_D_ ^d^ (nM)	K_D_ (nM)	K_D_ (nM)
hTNF	31.78 ± 0.25	0.751 ± 0.02	0.23	0.41	0.28
hCCL25	5.54 ± 0.35	3.74 ± 0.1.8	6.75	4.86	-
hCXCL12	1.66 ± 0.1	8.25 ± 0.2	49.76	16.6	-
hCXCL13	0.72 ± 0.04	1.55 ± 0.0^8^	21.40	13.2	-

^a^ Calculated affinity constants for the indicated proteins, as reported in [22]. ^b^ K_a_, association rate constant; ^c^ k_d_, dissociation rate constant; ^d^ K_D_, equilibrium dissociation constant.

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
