# Peer review of "Addition of a Viral Immunomodulatory Domain to Etanercept Generates a Bifunctional Chemokine and TNF Inhibitor"

_jcm, 2019, doi:10.3390/jcm9010025_

Round 1

Reviewer 1 Report

The authors present a manuscript that shows the results of testing an interesting idea:  Given that viruses product CRM proteins that have dual functionality (anti-TNF and chemokine binding), then what would happen if they took an effective FDA-approved anti-TNF molecule and combined it with a chemokine binding domain?  Potentially, the result would be even better than with the original anti-TNF drug. 

The authors made a variety of fusion proteins in insect cells, with the TNFR now linked to a chemokine binding domain (and also fused to an Fc domain, since the drug includes that). The variety is derived from trying various SECRET domains (the chemokine binding part) from CRM proteins.  Their most successful construct used stand-alone chemokine binding proteins SCP1 and SCP-3 (SECRET-containing protein), since for some reason the chimeras from SECRET domains that came from larger multi-functional proteins did not show any function. 

The authors show that their TNF-SCP proteins do bind TNF, do inhibit CCL25-induced chemotaxis, and TNF-SCP3 does alleviate clinical signs of rheumatoid arthritis in mice, in approximately the same amount as the anti-TNF FDA-approved drug.

So overall, the paper is a success and worthy of publication.  They do not necessarily show that they have improved the drug, but they had indeed added some functionality and this can be developed further in the future. 

I would like to suggest some changes:

It would be useful to have the authors discuss which chemokines their proteins should bind, and relate that to which chemokines are believed to be involved in rheumatoid arthritis.  In particular, they have a fairly successful SCP-3 construct, but the reader has to go to a previous paper to see which chemokines it binds, and the link is never made with RA. Their constructs all involve dimerization by the Fc domains (I think). But SECRET domains and SCP proteins are monomers, correct? Would the authors please discuss structurally how these proteins are believed to function in their construct? Wouldn’t the Fc domain pull the SECRET domains too close together?  The authors do mention that the domains can be pulled out separately to function (line 187), but where would the expected N- and C-termini be for the SECRET domains, and would having a huge protein at both ends be expected to hinder function?   How many disulfides are in the construct? The authors appear to be lacking a control in Figure 2, where they examine TNF-induced cytotoxicity. They show that some of their constructs inhibit the cytotoxicity, which is a good result.  But where is the hTNFR or the hTNFR-Fc control?  I don’t think they necessarily need to go back and do the experiments, but perhaps they could mention that they (probably) have done such experiments many times, and the result is that of course hTNFR does inhibit TNF cytotoxicity in those cells.

Smaller issues:

The sentence from 69-75 is confusing an seems grammatically difficult.  Can this be re-worded please? I would like to see references for the sentence ending on line 51, and the sentence ending on line 63. Figure 1: I can’t quite figure out what the bottom construct actually is (line 208, green part).  Is this just CRMD-Fc?  Then what are the purple ovals compared to the green rectangle? In the Table 2 headings, my preferences might be old-school, but I would say: the correct format to show an on rate and an off rate is to use lower case “k”, rather than upper case “K”. The lower case rates are used to derive the upper case dissociation constant KD.  So it would be more correct to use konand koffto denote the individual rate constants.

Author Response

Comments and Suggestions for Authors
The authors present a manuscript that shows the results of testing an interesting idea:Given that viruses product CRM proteins that have dual functionality (anti-TNF and chemokine binding), then what would happen if they took an effective FDA-approved anti-TNF molecule and combined it with a chemokine binding domain? Potentially, the result would be even better than with the original anti-TNF drug. The authors made a variety of fusion proteins in insect cells, with the TNFR now linked to a chemokine binding domain (and also fused to an Fc domain, since the drug includes that). The variety is derived from trying various SECRET domains (the chemokine binding part) from CRM proteins. Their most successful construct used stand-alone chemokine binding proteins SCP1 and SCP-3 (SECRET-containing protein), since for some reason the chimeras from SECRET domains that came from larger multifunctional proteins did not show any function. The authors show that their TNF-SCP proteins do bind TNF, do inhibit CCL25-induced chemotaxis, and TNF-SCP3 does alleviate clinical signs of rheumatoid arthritis in mice, in approximately the same amount as the anti-TNF FDA-approved drug. So overall, the paper is a success and worthy of publication. They do not necessarily show that they have improved the drug, but they had indeed added some functionality and this can be developed further in the future.

I would like to suggest some changes:
It would be useful to have the authors discuss which chemokines their proteins should bind, and relate that to which chemokines are believed to be involved in rheumatoid arthritis. In particular, they have a fairly successful SCP-3 construct, but the reader has to go to a previous paper to see which chemokines it binds, and the link is never made with RA.
Response:
In the Introduction section (lines 91-92), the group of chemokines bound by SCPs is already described. To facilitate the reader to find this information, it has been incorporated into the last section of the Discussion, which has also been extended to include further examples of SCP-targeted chemokines that are known to be involved in RA.
1. On line 524, "other chemokines directly targeted by the SCP3" has been completed and now reads "other chemokines directly targeted by the SCP3 (CCL20, CCL25, CCL28, CXCL12b, CXCL13, CXCL14, XCL1 and CX3CL1)"
2. On line 529 a new paragraph including several additional examples of SCP-targeted chemokines involved in RA development are included.
"CX3CL1 is involved in the migration of synovial fibroblasts and in angiogenesis and its blockade using specific antibodies was found to reduce inflammatory cell infiltration and bone erosion in a murine RA model (Ref.48). CCL20 may have an important role in RA development (Ref.49), and its blockade has been shown to hinder migration of Th17 cells to the inflamed joints in a murine model (Ref.50). Likewise, CCL25 mediated cell migration and production of inflammatory mediators may contribute to RA (Ref.51)."
Their constructs all involve dimerization by the Fc domains (I think). But SECRET
domains and SCP proteins are monomers, correct? Would the authors please discuss structurally how these proteins are believed to function in their construct? Wouldn’t the Fc domain pull the SECRET domains too close together? The authors do mention that the domains can be pulled out separately to function (line 187), but where would the expected N- and C-termini be for the SECRET domains, and would having a huge protein at both ends be expected to hinder function? How many disulfides are in the construct?
Response:
Etanercept (hTNFR2-Fc) does form Fc-mediated dimers and we expect that the fusion proteins purified here will form dimers, too, although higher order oligomers cannot be excluded. Our own data on the elution profiles of the hTNFR2_SCP3 on size exclusion chromatography suggest that it may form oligomers, although we have not assessed their precise composition. There is no published evidence about the oligomerization (or not) of the viral SCPs or CrmD/CrmB molecules to our knowledge. One report (Gileva et al. Properties of the recombinant TNF-binding proteins from variola, monkeypox, and cowpox viruses are different. Biochim Biophys Acta. 1764(11): 1710-8. 2006) showed that among recombinant CrmD / CrmB proteins from different viruses some
may be dimeric while others are monomers. Finally, we have shown before and in this report that the recombinant CrmD-Fc fusion protein, which has the same domain architecture in terms of N-to C-terminal primary sequence as the fusion proteins tested in this manuscript, can bind and block the activity of both TNF and CK ligands. Therefore, we believe that, although the CrmD-derived SECRET domain which was crystallized can bind to chemokines in its monomeric form, Fc mediated dimerization of the full length molecules containing SECRET domains does not necessarily preclude their activity.
Of course, it will be of interest to determine both the oligomerization status as well as the binding stoichiometry to their different ligands of these proteins in order to better understand their mechanisms of action.

To address these issued we have made the following modifications to the manuscript:
1. To clarify the domain architecture in the constructs, the following text was added in line 441:
"This was done maintaining the same domain architecture as present on the viral bifunctional receptors CrmD and CrmB, i.e. an N-terminal TNF binding domain followed by a CK-binding domain. The IgG1 Fc portion was located at the C-terminal end of the fusion molecules, as in etanercept. "
2. To emphasize the potential relevance of oligomerization status of the dual TNF/CK inhibitors on their activity we have included in line 459 the sentence
"While the SECRET domain is thought to function as a monomer (Ref.18), Fc fusion proteins are known to tend to form dimers, raising the question as to how this can possibly affect interaction with CKs." and in line 464 the sentence:
"The oligomerization status of these molecules as well as the binding stoichiometries to their different ligands remain to be addressed."
The authors appear to be lacking a control in Figure 2, where they examine TNFinduced cytotoxicity. They show that some of their constructs inhibit the cytotoxicity, which is a good result. But where is the hTNFR or the hTNFR-Fc control? I don’t think they necessarily need to go back and do the experiments, but perhaps they could mention that they (probably) have done such experiments many times, and the result is that of course hTNFR does inhibit TNF cytotoxicity in those cells.
Response:
As mentioned, we have extensively used this assay before with different recombinant proteins including in-house purified hTNFR2. This information has been included by adding in line 275 the following sentence with the corresponding reference:
1. "To do so, we employed a hTNF-mediated cytotoxicity assay previously established in our laboratory in which cell death can be efficiently blocked by hTNFR2 (Ref.15).
2. Also, hTNFR2 is used as a control group in similar experiments shown in Figure 3, showing its blocking activity.

Smaller issues:
The sentence from 69-75 is confusing an seems grammatically difficult. Can this be reworded please?
Response:
This sentence has been modified and now reads as follows:
1. "Currently, five different inhibitors of TNF are in use. Infliximab, adalimumab, and golimumab are monoclonal antibodies against TNF and certolizumab is a PEGylated anti-TNF Fab' fragment, while etanercept is a recombinant fusion protein including the extracellular domain of human TNFR2 receptor fused to the human IgG1Fc domain."

I would like to see references for the sentence ending on line 51, and the sentence ending on line 63.
Response: The reference Smith et al. 2013 has been added to the first sentence.

It is not clear which sentence ending line 63 the reviewer refers to. All sentences around line 63 have a reference at the end.
Figure 1: I can’t quite figure out what the bottom construct actually is (line 208, green part). Is this just CRMD-Fc? Then what are the purple ovals compared to the green rectangle?
Response: Figure 1 and its legend have been slightly modified to address these issues.

In the Table 2 headings, my preferences might be old-school, but I would say: the correct format to show an on rate and an off rate is to use lower case “k”, rather than upper case “K”. The lower case rates are used to derive the upper case dissociation constant KD. So it would be more correct to use konand koffto denote the individual rate constants.
Response: This mistake on our part has been corrected.

Reviewer 2 Report

Alejo et al have studied the engineering of an already available clinical immune response modifier (commerical TNF blockade). They propose the use of a poxvirus modifier as a potential way to improve the clinical responses. The authors engineer a number of constructs with either full poxvirus protein or putative active domains. They use a TNF cytotoxicity assay to show that the construct can block the TNF activity while migration assays are used to prove chemokine binding activity. Following this selection process, they subsequently use a mouse model of OA to verify that the selected construct has in vivo effects. 

I really only have minor comments for the authors. Mostly related to statistics and making the article easier to read for colleagues. 

Figure 2: You should include some statistical analysis here for the TNFR2_SCP1-3 blocking effect, comparing to the TNF/no TNF control. Also within protein comparisons might be useful. 

Figure 3: Statistics for the dose curves. 

Figure 3/4: You switch from ng/mL to nM. I think you need to make it clear in the text what the nM dose is equivalent to? Is it  So does the nM chemokine dose far exceed the effect of blocking TNF at the top ng/mL dose in the cytotoxicity assay? Some indicator of equivalence would be useful for the reader?

The plasmon resonance studies - did you include a construct that lacked TNF blocking activity. As suggested from your cytotoxicity assay this may have been due to the structure of the recombinant protein and confirming this in plasmon resonance would be nice but not essential. 

The in vivo studies are very useful. They show an in vivo efficacy. However,with statistical analysis I would question their inclusion. The key goal here is to show that your construct is more efficient at reducing disease than commerically available ENBREL. Some form of 2-way anova would be good to achieving this. 

Finally was there any cellular analysis to verify a reduction in immune responses underlying the disease?

Author Response

Comments and Suggestions for Authors
Alejo et al have studied the engineering of an already available clinical immune
response modifier (commerical TNF blockade). They propose the use of a poxvirus modifier as a potential way to improve the clinical responses. The authors engineer a number of constructs with either full poxvirus protein or putative active domains. They use a TNF cytotoxicity assay to show that the construct can block the TNF activity while migration assays are used to prove chemokine binding activity. Following this selection process, they subsequently use a mouse model of OA to verify that the selected construct has in vivo effects.

I really only have minor comments for the authors. Mostly related to statistics and making the article easier to read for colleagues.
Figure 2: You should include some statistical analysis here for the TNFR2_SCP1-3 blocking effect, comparing to the TNF/no TNF control. Also within protein comparisons might be useful.
Response:
A multiple t-test comparison comparing to the TNF only group has been performed.
1. Significant differences (p<0.01) are indicated with asterisks on the revised Figure 2 and this is now indicated in the Figure legend.
2. Additionally, a line describing statistical analyses has been included in the
corresponding Materials and Methods section.

Figure 3: Statistics for the dose curves.
Response:
For all panels on Figure 3, multiple t-test comparisons of tested proteins against the control hTNFR2 group have been performed.
1. Significant differences (p<0.01) are indicated with asterisks on the revised Figure 3 and this fact is now indicated in the Figure legend.
2. Additionally, a line describing statistical analyses has been included in the
corresponding Methods section.

Figure 3/4: You switch from ng/mL to nM. I think you need to make it clear in the text what the nM dose is equivalent to? Is it So does the nM chemokine dose far exceed the effect of blocking TNF at the top ng/mL dose in the cytotoxicity assay? Some indicator of equivalence would be useful for the reader?
Response:
The cytokines hTNF and mTNF are used at a 0.6-1.2 nM concentrations depending on the assay. Complete blockade of hTNF by hTNFR2 or hTNFR2_SCP2/hTNFR2_SCP3 is detected in the presence of 15-fold (100ng added protein) or 28-fold (250ng added protein) molar excesses, respectively. In the case of mTNF, approximately ten times higher doses are required for complete blockade. To make the comparison with the migration assay easier, we have included the following modifications in the text:
1. In the methods section, we have modified line 163, "were incubated with 20 ng/ml of hTNF or 10-20 ng/ml TNF" for "were incubated with 20 ng/ml (1.2 nM) of hTNF or either 10 or 20 ng/ml (1.2 nM or 2.4 nM) of mTNF"
2. We have modified lines 302-303: "Thus, while 100 ng of hTNFR2 were sufficient to provide full protection against hTNF induced cytotoxicity" for "Thus, while 100 ng of hTNFR2 (corresponding to a molar excess of approximately 15 fold) were sufficient to provide full protection against hTNF induced cytotoxicity" .
3. On line 321: we have added the text "achieving complete blockade at 10-20 fold molar excess over the chemokine."
4. We have modified the legend to Figure 2: "L929 cells were incubated with 1.2 nM hTNF for 16 h to induce cell death. The hTNF used was preincubated with10, 50 or 100ng of each recombinant protein as indicated" for "L929 cells were incubated with hTNF to induce cell death. The hTNF used was preincubated with increasing doses of each recombinant protein as indicated"

The plasmon resonance studies - did you include a construct that lacked TNF blocking activity. As suggested from your cytotoxicity assay this may have been due to the structure of the recombinant protein and confirming this in plasmon resonance would be nice but not essential.
Response:
This has not been assessed. To make clearer that we don´t know how or if the binding to TNF of these molecules is affected (as opposed to its TNF blocking capacity), we have modified the text in the Discussion on lines 447-449 to read as follows: "Fusion of the hTNFR2 to SECRET domains from CrmB was possible, although the purified recombinant proteins showed a compromised TNF inhibitory activity. "

The in vivo studies are very useful. They show an in vivo efficacy. However,with
statistical analysis I would question their inclusion. The key goal here is to show that your construct is more efficient at reducing disease than commerically available ENBREL. Some form of 2-way anova would be good to achieving this.
Response:
We have analysed the data using both 2-way ANOVA, showing significant effects of time and group variables as well as multiple t-tests to compare individual groups to one another. The treatment with either ENBREL 100μg or hTNFR2_SCP3 50μg was found to significantly (p<0.01) reduce the signs of illness from days 28 to 36 (final time point) of the experiment. While the lower dose ENBREL 50μg or CrmD 50μg treatment regimes did show lower mean clinical scores, the differences to the control untreated group was only significant on days 28 or 28 and 30 of the experiment, respectively.
Moreover, significant differences between the ENBREL 50μg and ENBREL 100μg
groups were identified (days 28-36), showing a dose effect of ENBREL treatment as expected. The treatment with 50μg of hTNFR2_SCP3 showed no significant differences to the ENBREL100μg treatment group at any time point, while it was found to be significantly different from the ENBREL 50μg treatment group (days 28-36). In our interpretation, this suggests an increased efficacy of the bifunctional fusion protein over the hTNFR2 in this assay. In Figure 5B, a similar analysis showed statistically significant (p<0.01) differences of the groups treated with ENBREL or hTNFR_SCP3 as compared to the untreated groups from day 28 or day 30 on, respectively. No significant differences were found in the case of in-house produced hTNFR2 due to the limited number of animals in this group. As before, no significant differences are found between the ENBREL 100μg and the hTNFR_SCP3 50μg group, suggesting again an increased efficacy of the latter in this model. This reasoning is incorporated into the revised version of the manuscript as follows:
1. The text "The asterisk and bar on panel A indicate time points at which statistically significant (p< 0.01) differences between the ENBREL 50μg and hTNFR_SCP3 50μg treatment groups. In panel B, asterisks and bars indicate significant differences detected between the ENBREL 100μg (upper bar) or the hTNFR_SCP3 50μg (lower bar) as compared to the untreated control group. " has been added at the end of the legend to Figure 5.
2. The text at the end of the Results section (from line 380) has been modified to include the statistical data and ensuing reasoning.
3. A line describing statistical analysis has been added to the corresponding Methods section.

Finally was there any cellular analysis to verify a reduction in immune responses
underlying the disease?
Response: We did not carry out analysis of the cellular immune response under
different conditions. We hope this will be done in future studies.
